# Excitation of coupled spin–orbit dynamics in cobalt oxide by femtosecond laser pulses

Takuya Satoh [1,2], Ryugo Iida[2], Takuya Higuchi[3], Yasuhiro Fujii [4], Akitoshi Koreeda[4], Hiroaki Ueda[5], Tsutomu Shimura[2], Kazuo Kuroda[2], V.I. Butrim[6] & B.A. Ivanov[7,8]

Ultrafast control of magnets using femtosecond light pulses attracts interest regarding applications and fundamental physics of magnetism. Antiferromagnets are promising materials with magnon frequencies extending into the terahertz range. Visible or near-infrared light interacts mainly with the electronic orbital angular momentum. In many magnets, however, in particular with iron-group ions, the orbital momentum is almost quenched by the crystal field. Thus, the interaction of magnons with light is hampered, because it is only mediated by weak unquenching of the orbital momentum by spin–orbit interactions. Here we report all-optical excitation of magnons with frequencies up to 9 THz in antiferromagnetic CoO with an unquenched orbital momentum. In CoO, magnon modes are coupled oscillations of spin and orbital momenta with comparable amplitudes. We demonstrate excitations of magnon modes by directly coupling light with electronic orbital angular momentum, providing possibilities to develop magneto-optical devices operating at several terahertz with high output-to-input ratio.

[1] Department of Physics, Kyushu University, Fukuoka 819-0395, Japan. [2] Institute of Industrial Science, The University of Tokyo, Tokyo 153-8505, Japan. [3] Department of Physics, Friedrich-Alexander-Universität Erlangen-Nürnberg (FAU), 91058 Erlangen, Germany. [4] Department of Physical Sciences, Ritsumeikan University, Shiga 525-8577, Japan. [5] Department of Chemistry, Kyoto University, Kyoto 606-8502, Japan. [6] National University of Science and Technology "MISiS", Moscow 119049, Russia. [7] Institute of Magnetism, Ukrainian Academy of Science, 03142 Kiev, Ukraine. [8] Taras Shevchenko National University of Kiev, 03127 Kiev, Ukraine. Takuya Satoh and Ryugo Iida contributed equally to this work. Correspondence and requests for materials should be addressed to T.S. (email: satoh@phys.kyushu-u.ac.jp)

The inverse effect of the magneto-optical Faraday effect, specifically light acting on magnetic systems, was first attempted by Faraday in 1845[1] and verified more than 100 years later[2, 3]. Nowadays, the various inverse magneto-optical effects (the inverse Faraday effect (IFE) or the inverse Cotton–Mouton effect (ICME))[4] are used for non-thermal optical excitation of spin oscillations with frequencies in the terahertz range in transparent antiferromagnets[5–7]. The effects enable the control of spin dynamics using optical polarization. In contrast to a recent realization of a non-trivial spin evolution for opaque metallic ferrimagnets[8, 9], non-thermal excitation does not lead to intense heating of the sample. This feature attracts particular attention for possible applications of antiferromagnets in magnetic recording and magneto-optical devices, e.g., terahertz radiation sources[10, 11] and optomagnonic read–write transfer[12].

The coupled spin–orbit dynamics of cobalt oxide CoO has been studied for half a century. For CoO, quenching of the orbital angular momentum of a free $Co^{2+}$ ion ($L_{free} = 3$) by the cubic crystal field is only partial, resulting in an effective angular momentum of $L = 1$ (orbital triplet), which should be treated as an additional degree of freedom in the magnetic subsystem of CoO. Being subject to a low-symmetry crystal field, the ions with partial unquenching increase magnetic anisotropy (see ref. [13] and Supplementary Note 1); its magnitude is comparable to those of the effective spin–orbit and exchange interactions. All these specific features lead to magnon modes originating from spin and orbital degrees of freedom and the magnon frequencies are much higher than those for standard antiferromagnets containing transition-metal ions[14]. The magnon modes of CoO have been investigated using Raman scattering[15], infrared absorption[13, 16], infrared reflection[17] and inelastic neutron scattering[18, 19], but

their interpretation and theoretical description are still being debated. Further, the coupling of femtosecond laser pulses and unquenched orbital angular momentum has never been explored.

In the following, we report highly efficient non-thermal coherent excitation of magnon modes in CoO using femtosecond laser pulses at frequencies up to 9 THz. Symmetry analysis and a theoretical model calculation confirm the excitation of these modes, which consist of coupled dynamics that have comparable amplitudes of the oscillations of spin and orbital angular momenta.

## Results

**Analysis of magnon modes in CoO.** Below the Néel temperature $T_N = 292$ K, CoO exhibits an antiferromagnetic order with the antiparallel spin orientations of $Co^{2+}$ ions belonging to two sublattices (labeled with 1 and 2), $S_1$ and $S_2$ (effective $S = 3/2$). The unquenched part of the orbital angular momenta (effective $L = 1$) of $Co^{2+}$ ions, $L_1$ and $L_2$ are expected to be antiparallel to the corresponding spin momenta for any sublattice because of the spin–orbit interaction. For the crystallographic and magnetic structures (Fig. 1a), CoO exhibits a low-symmetry monoclinic ground state (crystal point group $2/m$) with spin directions inclined from the crystalline axis [001] by an angle $\rho$. The value $\sin \rho = \sqrt{2/51}$ is commonly accepted now[20]. We choose a coordinate system with the z-axis along this direction and $\hat{y} || [\bar{1}10]$.

In CoO, the coupled spin–orbit dynamics provides a complex combination of magnon modes with different symmetries and frequencies. We introduce convenient combinations of the variables, total spin angular momentum $\mathbf{m}_S = \mathbf{S}_1 + \mathbf{S}_2$, total orbital angular momentum $\mathbf{m}_L = \mathbf{L}_1 + \mathbf{L}_2$ (in the ground

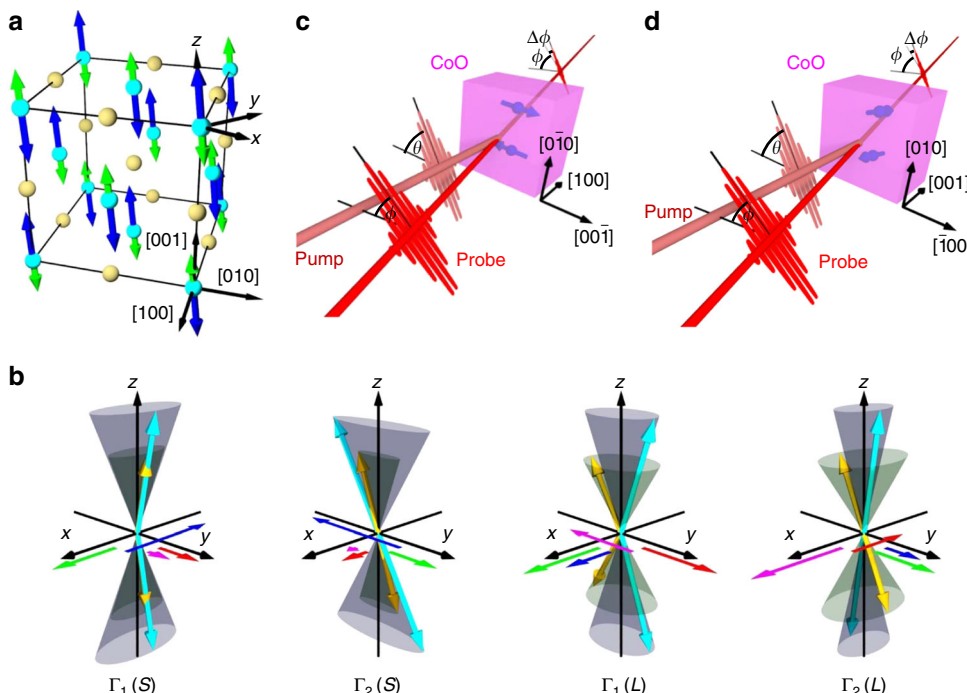

**Fig. 1** Experimental geometries and excited magnon modes. **a** Crystallographic and magnetic structures of CoO in the coordinate system. Cobalt atoms are shown as *cyan spheres*. *Blue* and *green arrows* represent spin and orbital angular momenta, respectively. Oxygen atoms are shown as *yellow spheres*. **b** Illustration of the four transverse magnon modes of CoO: $\Gamma_1(S)$, $\Gamma_2(S)$, $\Gamma_1(L)$ and $\Gamma_2(L)$. Spin ($\mathbf{S}_1$, $\mathbf{S}_2$) and orbital ($\mathbf{L}_1$, $\mathbf{L}_2$) angular momenta are represented as *cyan* and *yellow arrows*, respectively. $\mathbf{m}_S = \mathbf{S}_1 + \mathbf{S}_2$, $\mathbf{m}_L = \mathbf{L}_1 + \mathbf{L}_2$, $\mathbf{n}_S = \mathbf{S}_1 - \mathbf{S}_2 - N_S\hat{\mathbf{z}}$ and $\mathbf{n}_L = \mathbf{L}_1 - \mathbf{L}_2 - N_L\hat{\mathbf{z}}$ are shown as *red, magenta, blue* and *green arrows*, respectively. The ratio of the amplitude of the variables $(m_S)$, $(m_L)$, $(n_S)$ and $(n_L)$ are $\Gamma_1(L)$: $(m_L)_y/(m_S)_y = -1.02$, $(n_L)_x/(n_S)_x = 1.17$; $(n_S)_x/(m_S)_y = -0.62$, $\Gamma_1(S)$: $(m_L)_y/(m_S)_y = -0.57,(n_L)_x/(n_S)_x = -0.65$; $(n_S)_x/(m_S)_y = -2.36$, $\Gamma_2(L)$: $(n_L)_y/(n_S)_y = = 1.65$; $(m_L)_x/(m_S)_x = -1.50$, $(n_S)_x/(n_S)_y = -1.61$, $\Gamma_2(S)$: $(n_L)_y/(n_S)_y = -0.44$; $(m_L)_x/(m_S)_x = 0.40$, $(n_S)_x/(n_S)_y = -0.25$. **c** Transverse and (**d**) longitudinal geometries. $\theta$ and $\phi$ denote the azimuthal angles of the pump and probe polarizations from the reference axes, which are [001] and [100] in transverse and longitudinal geometries, respectively

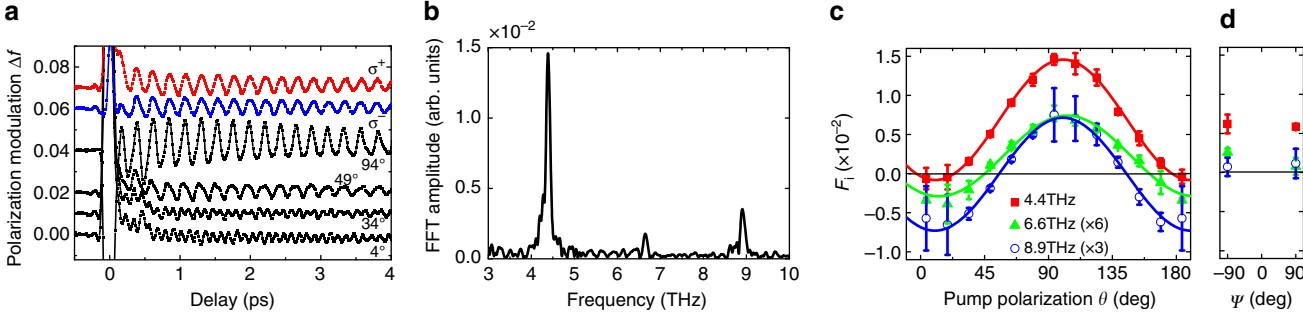

**Fig. 2** Terahertz magnon modes in the transverse geometry. **a** Change $\Delta f$ in probe-light polarization as a function for the delay time from the pump light to the probe light at $T = 5$ K in the transverse geometry. Pump pulses were linearly polarized with different values of $\theta$ or circularly polarized $\sigma^{\pm}$ ($\psi = \mp 90°$). **b** Fourier-transformed amplitude spectrum of the result with $\theta = 94°$ in (**a**). **c**, **d** Pump polarization dependence of the signed amplitude $F$ in the 4.4, 6.6 and 8.9 THz modes fitted by function $\Delta f(t) \equiv Fe^{-\alpha \Omega t} \sin(\Omega t + \vartheta)$. The fitted results of $F$ at 4.4 THz (*solid line*), 6.6 THz (*dotted line*) and 8.9 THz (*dashed line*) are shown as functions of $\theta$ for linear polarizations (**c**) and of $\psi$ for circular polarizations (**d**). The error bars represent the SD of the measurements

state $\mathbf{m}_S = \mathbf{m}_L = 0$), and spin and orbital antiferromagnetic vectors, $\mathbf{N}_S = \mathbf{S}_1 - \mathbf{S}_2$ and $\mathbf{N}_L = \mathbf{L}_1 - \mathbf{L}_2$, respectively. The antiferromagnetic vectors can be present through their values in the ground state $N_{S,L}\hat{\mathbf{z}}$ and the small deviations from the ground state, $\mathbf{n}_{S,L} \perp \hat{\mathbf{z}}$. Our theoretical analysis (see Supplementary Note 1 and 2) suggests that four transverse magnon modes should be observed, which are classified into two modes with different symmetries ($\Gamma_1$ and $\Gamma_2$). These magnon modes (Fig. 1b) belong to two symmetry classes: (1) $\Gamma_1(S)$ and $\Gamma_1(L)$-modes with nonzero $(n_S)_x$, $(n_L)_x$, $(m_S)_y$ and $(m_L)_y$; and (2) $\Gamma_2(S)$ and $\Gamma_2(L)$ modes with nonzero $(n_S)_y$, $(n_L)_y$, $(m_S)_x$ and $(m_L)_x$[14]. Here, (S) and (L) denote spin- and orbital-dominated modes, respectively. They should exhibit different polarization dependence in the magneto-optical experiments.

**Experimental geometries**. To demonstrate the coherent excitation of magnons in CoO and to investigate these complex dynamics and symmetries of the magnons, we performed time-resolved pump–probe experiments. Here, optical pulses excite the magnons through the IFE and the ICME. In particular, we carefully chose the crystalline orientation and the optical polarizations to distinguish magnon modes with different symmetries. In the transverse and longitudinal geometries (TG and LG, respectively), the pump beam propagates along the [100] and [001] directions, which are nearly perpendicular and parallel to the $z$ axis (Fig. 1c, d, respectively). The electric field of the pump light has the form $E_i(t) = \mathrm{Re}[\mathscr{E}_i(t)e^{i\omega t}]$. The time-dependent complex amplitude of the electric field $\mathscr{E}_i(t)$ takes the form $\mathscr{E}_{[001]}(t) \equiv \mathscr{E}_0(t)\cos\theta$, $\mathscr{E}_{[0\bar{1}0]}(t) \equiv \mathscr{E}_0(t)\sin\theta e^{i\psi}$ in the TG, and $\mathscr{E}_{[100]}(t) \equiv \mathscr{E}_0(t)\cos\theta$, $\mathscr{E}_{[010]}(t) \equiv \mathscr{E}_0(t)\sin\theta e^{i\psi}$ in the LG, where $0° \leq \theta < 180°$, $-90° \leq \psi \leq 90°$. For linearly polarized light, $\psi = 0$ with angle $\theta$ determining the azimuth of the polarization; for circularly polarized light, $\theta = 45°$ and $\psi = \mp 90°$ determine the two different helicities $\sigma^{\pm}$. For both geometries, pump pulses were circularly polarized ($\sigma^{\pm}$) or linearly polarized with different values of $\theta$. For experimental details, see Methods.

**Magnon excitation in the TG**. Figure 2a expresses the change $\Delta f$ in probe polarization, see Methods for definition, as a function of delay time $t$ in the TG. Figure 2b gives the Fourier-transformed amplitude spectra of the oscillations for $\theta = 94°$ in Fig. 2a. Excitations with frequencies of 4.4, 6.6 and 8.9 THz are clearly observed. These modes have been attributed to magnetic excitations[15].

The temporal evolutions of the probe polarization observed in the TG were fitted by a superposition of damped oscillations $\Delta f(t) \equiv \sum_{j=1}^{3} F_j e^{-\alpha_j \Omega_j t} \sin(\Omega_j t + \vartheta_j)$ with three

frequencies $\Omega_j/2\pi = 4.4$, 6.6 and 8.9 THz, and damping constants $\alpha_j = 0.011 \pm 0.001$, $0.004 \pm 0.002$ and $0.009 \pm 0.003$ for $j = 1, 2$ and 3, respectively. Here, $F_j$ is defined as the signed amplitude that may take negative values. Figure 2c, d shows the signed amplitude $F_j$ for linear and circular polarizations of pump, respectively. From Fig. 2c, the dependence of the pump polarization is almost proportional to $(1 - \cos 2\theta)$ for the 4.4-THz mode, $\cos 2\theta$ for the 8.9-THz mode and ($\cos 2\theta +$ const.) for the 6.6-THz mode, where const. is neither 0 nor $-1$. Surprisingly, for a circularly polarized pump beam, the amplitudes were almost independent of helicity $\sigma^{\pm}$ and nearly equal to that for a linearly polarized pump beam with $\theta = 45°$, 135°.

To explain this complicated picture, let us consider the phenomenological theory (see Supplementary Note 3 for detail). The (inverse) magneto-optical effects result from the dependence of the permittivity on the magnetic order parameter[21]. Namely,

$$\delta \varepsilon_{ij} = ik_{ijk}m_k + 2g_{ijkl}N_k n_l, \tag{1}$$

where $k_{ijk}$ and $g_{ijkl}$ determine the Faraday effect (FE) and the Cotton–Mouton effect (CME) (both direct and inverse), respectively. Here, $k_{ijk} = -k_{jik}$, $g_{ijkl} = g_{jikl} = g_{ijlk}$, the structures of these tensors are determined by the magnetic symmetry group of the crystal[22, 23]. It is noteworthy that the subscript "$l$" is not related to the orbital angular momentum, but a lower index for a tensor "$ijkl$". For materials such as CoO with partially unquenched orbital angular momentum, the independent dynamics of the spin and orbital angular momenta should be considered. Formally, the independent contributions of these momenta, $\mathbf{m}_S$, $\mathbf{n}_S$ or $\mathbf{m}_L$, $\mathbf{n}_L$ can be written down in the form given in Eq. (1). The symmetric properties of the oscillations of the corresponding spin and orbital vectors are indeed the same, as well as the symmetric properties of the tensors $k_{ijk}$ and $g_{ijkl}$. Although both the orbital and spin angular momenta contribute to the magnetism in CoO, the magneto-optical effect is dominated by the orbital angular momentum because the optical selection rule for the electric-dipole interaction only allows changes in orbital angular momentum. A change in spin can be only allowed via spin–orbit interactions. In many other systems, however, the orbital angular momentum is quenched and cannot contribute to the magnetic properties of media, and thus magneto-optical effect is governed by the indirect coupling between spin and light[24]. To simplify the expressions, we omit the index "$L$" from vectors $\mathbf{m}_L$, $\mathbf{n}_L$ in the following. The effective energy of the magneto-optical interaction with the use of Eq. (1)

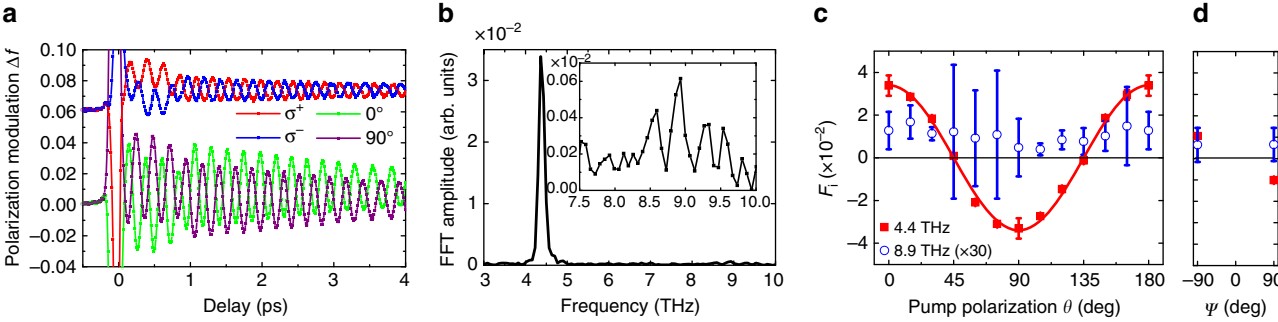

**Fig. 3** Terahertz magnon modes in the longitudinal geometry. **a** Change $\Delta f$ in the probe-light polarization as a function of delay time from the pump light to the probe light at $T = 5$ K in the longitudinal geometry. Pump pulses were linearly polarized with different values of $\theta$ or circularly polarized $\sigma^{\pm}$ ($\psi = \mp 90°$). **b** Fourier-transformed amplitude spectrum of the result with $\theta = 0°$ in (**a**). The inset shows a magnification of the spectrum. **c**, **d** Pump polarization dependence of the signed amplitude $F$ in the 4.4 and 8.9 THz modes fitted by function $\Delta f(t) \equiv F e^{-\alpha \Omega t} \sin(\Omega t + \vartheta)$. The fitted results of $F$ at 4.4 THz (*solid line*) and 8.9 THz (*dotted line*) are shown as functions of $\theta$ for linear polarizations (**c**) and of $\psi$ for circular polarizations (**d**). The error bars represent the SD of the measurements

for TG can be written as:

$$W_{\text{TG}}^{\text{MO}}(t) = -\frac{I(t)}{16\pi}[(G_1 + G_2 + G_3)n_x + (G_1 - G_2 - G_3)$$
$$n_x \cos 2\theta - G_4 n_y (1 - \cos 2\theta)$$
$$+ (G_5 n_x + G_6 n_y)\sin 2\theta \cos \psi$$
$$+ (K_1 m_y - K_2 m_x)\sin 2\theta \sin \psi], \quad (2)$$

where $I(t) = \mathscr{E}(t)\mathscr{E}^*(t)$ for a given polarization of light. Here, $K$s and $G$s are related to the FE and CME (both direct and inverse), respectively, and are defined in Supplementary Eqs. (20)–(28).

When a material has high symmetry, such as cubic symmetry, the parameters $G_1$, $G_2$, $G_3$ and $G_4$ are zero and the amplitude of the magnetic oscillation induced by the ICME gives $\sin 2\theta \cos \psi$ dependence in Eq. (2)[12, 25–27]. This implies that its sign changes when the pump azimuthal angle $\theta$ is changed from 45° to 135° for linearly polarized light (i.e., $\psi = 0°$), and that it vanishes for circularly polarized light ($\psi = \pm 90°$). Until very recently, magnetic excitations with only this particular polarization dependence were considered to originate from the ICME. However, the magnon modes experimentally observed in the TG never exhibit standard $\sin 2\theta \cos\psi$ dependence. In a crystal with a reduced symmetry like in CoO, other $G$-related terms, $G_1$–$G_4$, can also appear in Eq. (2). These terms may lead to magnetic oscillations induced by the ICME that are independent of helicity ($\psi$) and proportional to $\cos 2\theta$ or $(1 - \cos 2\theta)$. It is noteworthy that the $(1 - \cos 2\theta)$ contribution from ICME does not vanish even for circularly polarized light with $\cos 2\theta = 0$ and the amplitude for circular polarization should be equal to that for linear polarization with $\theta = 45°$ and 135°, as mentioned above. This relationship enables other mechanisms to be excluded, e.g., thermal excitation. This helicity-independent excitation of the magnon modes by circularly polarized light is a non-standard manifestation of the ICME and is a characteristic of materials with low symmetry[28].

**Magnon excitation in the LG.** Figure 3a shows $\Delta f$ as a function of $t$ in the LG. The amplitude of oscillations in $\Delta f$ reaches large values, as high as 0.02. The output-to-input ratio, which is defined as the amplitude of oscillation in $\Delta f$ normalized by the pump fluence and pump–probe spectral weight is two orders of magnitudes higher than that observed in NiO[28] (see Supplementary Note 4). It is noteworthy that $\Delta f$ was measured under non-resonance conditions for both CoO and NiO. This demonstrates the high efficiency of the optical excitation for magnets with unquenched orbital angular momentum. Figure 3b presents the Fourier-transformed amplitude spectrum of the

oscillations for $\theta = 0°$ in Fig. 3a. The temperature dependence of the spectral magnitude further confirms that the 4.4-THz spectral peak is of magnetic origin (see Supplementary Fig. 1). Aside from the 4.4-THz mode, a small peak at 8.9 THz was found. Figure 3c, d show the signed amplitude $F_j$ for linear and circular pump polarizations, respectively. $F_i$ are almost proportional to $\cos 2\theta$ for the 4.4-THz and nearly constant for the 8.9-THz modes. In contrast to the TG, the sign of $F$ for the 4.4-THz mode changes by reversing the pump helicity, clearly identifying the IFE[28–32].

**Discussion**

We now discuss the symmetry of the magnon modes. Our analysis (see Supplementary Note 3) reveals that the 4.4-THz mode is the $\Gamma_2$ mode, which is excited with the $(1 - \cos 2\theta)$-dependent ICME in the TG (Supplementary Eq. (31)), with the $\cos 2\theta$-dependent ICME (Supplementary Eq. (35)) and with the helicity-dependent IFE (Supplementary Eq. (36)) in the LG. The polarization dependences of the 6.6-THz mode in the TG is neither $\cos 2\theta$ nor $(1 - \cos 2\theta)$ (Supplementary Eq. (29)) and hence this mode can be interpreted as a $\Gamma_1$ mode. If the high-frequency signal is caused by an excitation of one pure mode, either $\Gamma_1$ mode or $\Gamma_2$ mode, its $\theta$-dependence should follow one of the dependencies observed for low-frequency modes. From the experimental data, this is not evident for the TG. Thus, we suggest that the signal observed at 8.9 THz is a superposition of signals of two excited modes of different symmetries but similar frequencies. Our experimental results indicate that the lower-frequency (4.4 and 6.6 THz) modes with different symmetries have significantly different frequencies, whereas the higher-frequency (8.9 THz) modes with different symmetries are degenerate. This finding contrasts with previous studies in which the low-frequency $\Gamma_1$ and $\Gamma_2$ modes were claimed to be almost degenerate[13, 15], probably because the samples were not confirmed to be a single domain. Our spontaneous Raman scattering measurements on a single domain support our conclusion (see Supplementary Fig. 2).

Previous theoretical studies on CoO spin–orbit dynamics were based on the quasi-uniaxial models, where the magnetic anisotropy originates mainly from the crystal field having the form $-C\left(L_{z,1}^2 + L_{z,2}^2\right)$[13–19]. Such models obviously lead to almost-degenerate doublets of the modes with $\Gamma_1$ and $\Gamma_2$ symmetries, which is not consistent with our observations. To describe our experimental results, we propose a quantum mechanical model with a biaxial crystal field for states with orbital angular momentum $L = 1$ (see Supplementary Note 1 for details). For transverse oscillations, four modes of coupled spin–orbit

dynamics are found analytically and the parameters of the Hamiltonian are determined. The pair of lower-frequency modes appears to be spin dominated, whereas the almost-degenerate pair of higher-frequency modes is orbital dominated. It is noteworthy that the different degrees of lifting the degeneracy of the lower-frequency and higher-frequency modes require the notion of competing spin and orbital contributions to the anisotropy. See Supplementary Notes 1, 2 for more details.

Here we discuss the reason why the 4.4-THz mode was not excited by the IFE in the TG. In principle, this mode can be excited by the IFE because of the term $K_2$ in Supplementary Eq. (32), as well as the ICME from term $G_4$ in Supplementary Eq. (31). The reason why this mode was not excited by the IFE is the following. The 4.4-THz mode is a $\Gamma_2(S)$ mode, where the trajectory of the magnetization vector is an ellipse elongated along the $y$ axis. Excitation by the IFE initializes as a kick on the magnetization along the $y$ direction, whereas that by the ICME is along the $x$ direction, which corresponds to the short axis of the ellipse. Therefore, the ICME dominates the excitation efficiency and the helicity-dependent excitation via the IFE was not observed.

To summarize, we demonstrated that femtosecond laser pulses can efficiently excite magnons consisting of the spin and unquenched orbital angular momentum in CoO. The study of coupled spin–orbital dynamics is of great interest in the promising area of optomagnonics. First, it enables the realization of faster spin dynamics, in particular excitation of magnons with higher frequencies than those mediated by pure spin dynamics. Second, the materials with a large fraction of orbital angular momentum oscillations are expected to exhibit higher efficiency in their magneto-optical effects (either inverse or direct), which leads to higher amplitudes of both excited magnons and of the probe signal. Indeed, we found that CoO produces quite a high magneto-optical signal from the excited magnons even for frequencies of order 10 THz.

## Methods

**Sample**. The samples were CoO (001) biaxial single crystals[33] grown by the floating zone method. Magneto-striction leads to contraction of the cubic cell along the $\langle 100 \rangle$ direction and gives rise to three types of T domains[34]. From the TG- and the LG-configured pump–probe measurements, two types of T domains were classified from observations of differences in birefringence in the cross-Nicol configuration using a polarization microscope. In the TG, its optic axes were 4° out of alignment with the [010] and [001] directions, and $\Delta n \simeq 0.02$ for 633 nm at 5 K; in the LG, the crystal had axes in the [110] and [$1\bar{1}0$] directions, and $\Delta n \simeq 0.001$. The values agree with those taken from the literature[33, 34]. The thicknesses of the samples were 70 μm for the TG and 50 μm for the LG.

**Pump–probe experimental setup**. The samples were cooled at 5 K in a cryostat in the absence of an external magnetic field. A Ti: sapphire regenerative laser amplifier (Spectra-Physics, Spitfire Pro) was used as the fundamental light source producing a central wavelength of 800 nm, a pulse duration of 50 fs and a repetition frequency of 1 kHz. With an optical parametric amplifier, a part of this light was converted to a wavelength of 1,500 nm and used as pump pulses. The pump wavelength was chosen to avoid the real ($d$–$d$) excitation[35]. The pump–pulse fluence was 130 mJ cm$^{-2}$. The rest of the light passed through a delay line before entering the sample as time-delayed probe pulses. To obtain a maximal signal, in the TG, probe pulses were linearly polarized with $\phi = 26.5°$ from [001] that enabled the simultaneous detection of the FE, CME and linear dichroism, whereas in the LG the probe pulses were linearly polarized at $\phi = 45°$ (see Supplementary Fig. 3). The change in the probe polarization was obtained by measuring the balance of the two linearly polarized components from the transmitted probe light; these orthogonal components, $I_1$ and $I_2$ have the same magnitude. $I_1$ and $I_2$ were ±45° from the reference angle when injected into the sample. The transmitted probe pulse was divided into two orthogonally polarized pulses by a Wollaston prism, each pulse was detected using a Si photodiode. We calculated $f = (I_1 - I_2)/(I_1 + I_2)$ and regarded $\Delta f(t)$ as a modulation of the probe polarization[26].

**Data availability**. Data that support the findings of this study are available from the corresponding author on request.

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

## Acknowledgements

We thank H. Tamaru and K. Tomiyasu for valuable discussions, and K. Tsuchida for technical assistance. T. Satoh was supported by JST-PRESTO, JSPS KAKENHI (numbers JP15H05454 and JP26103004), JSPS Core-to-Core Program (A. Advanced Research Networks) and Kyushu University Short-term International Research Visitation Program. T.H. was supported by JSPS Postdoctoral Fellowship for Research Abroad and the European Research Council (Consolidator Grant NearFieldAtto). B.A.I. was supported by JSPS Invitation Fellowship Programs for Research in Japan and the National Academy of Sciences of Ukraine (No. 1/16-N). V.I.B. was supported financially by the Russian Foundation for Basic Research (No. 16-02-00069 a) and the Increase Competitiveness Program of NUST "MISiS" (Act 211 of the Russian Federation, contract number 02. A03.21.001).

## Author contributions

T. Satoh conceived the project. R.I. and T. Satoh carried out the pump–probe experiment. Y.F., A.K. and T. Satoh measured spontaneous Raman spectra. B.A.I. and V.I.B. performed the model calculation. B.A.I., V.I.B., R.I., T.H. and T. Satoh analyzed the data. H.U. provided the sample. T. Shimura and K.K. supervised the project. T. Satoh, R.I., T.H., B.A.I. and V.I.B. wrote the manuscript. All authors discussed the results and commented on the manuscript.

## Additional information

**Competing interests:** The authors declare no competing financial interests.

