## [Peer Review File · Nature Communications]

Reviewer #1 (Remarks to the Author):

The authors report on experimental study of laser-induced coherent dynamics of a magnetic order parameter in an antiferromagnet CoO. The feature of this compound is the unquenched orbital momentum, which is in contrast to iron oxides, extensively studied by femtosecond optical pump-probe methods in the recent years. The main finding reported in the manuscript is that four magnon modes are excited via inverse Faraday and Cotton-Mouton effects. The symmetries of the modes are determined from the pump-polarization dependences. By performing comprehensive analysis of the magnon frequencies the authors determine contributions from orbital and spin momenta to these modes.

The experimental demonstration of coherent dynamics of magnetic order parameter in the compound with unquenched orbital momentum, excited via ultrafast inverse magneto-optical effects is the novel result and of interest to the community working in the field of femtomagnetism. However, the extended theoretical analysis mainly concerns the frequencies of the modes, and the experimental data on excitation of these modes are required only to get the symmetry of the modes. In fact, Raman scattering experiments could yield the same information (Suppl. information, Section 6). Therefore, the manuscript in general does not provide insight into yet unexplored features of coupling of femtosecond pulses to compounds with unquenched orbital momentum. Thus, although the experimental observations and the theoretical analysis are of interest to the particular community of researchers, I cannot recommend publication of the manuscript in the present form in Nature Communications. Below I list the specific issues which the authors should address before the manuscript can be considered for publication.

1. Supplementary information. Parts 1-2. Presented theoretical analysis seems to be the key element of the manuscript. Therefore, the validity of the model, considered here, should be carefully substantiated. The authors introduce the Hamiltonian (S1), which includes distinct terms describing anisotropy for the orbital moments and for spins. It is generally accepted that the magneto-crystalline anisotropy originates from the spin-orbital coupling. Thus, it is not clear, what is the source of anisotropy for spins, if not the coupling to orbital moments. Could the authors give a brief explanation of the physics behind different terms in Eq. (S1). It is necessary, since the manuscript is meant for a broad research community.

2. The magnon mode of 8.9 THz in the LG geometry is extremely weak and its amplitude in the FFT spectrum (Fig. 3c) seems not to exceed the noise level. I suggest the authors to present more experimental data, proving that the 8.9 THz peak in the FFT spectra is a reliable one (e.g. data for the probe polarization of 135 deg, at which the amplitude of the 8.9 THz oscillations seems to be the largest (Fig. S2b)). How important the fact that this mode is observed for the conclusions of the manuscript?

3. Discussion of the results presented in Fig. 2c. The authors write that the polarization dependences of the amplitudes of the modes 6.6 and 8.9 THz are proportional to $(\cos(2\theta)+\text{const})$ and $\cos(2\theta)$, which contradicts the data in Fig. 2c. In fact, from this Figure one can see that the amplitudes of all three modes observed in the given experimental geometry can be described by a unified expression $(C-A*\cos(2\theta))$, where A and C take specific values for each mode.

4. It is not clear for me, why the 4.4 THz mode excited in the transversal geometry by circularly-polarized pump pulses is insensitive to the pump helicity. From Eq. (S1-S32) it follows that there should be a clear sign-change, if one takes into account strong ellipticity of the $\Gamma_2(S)$ mode.

5. When analyzing the coupling of light to the excited modes (Suppl. Material, section 3) the authors consider only the terms linear on dynamic components of orbital momentum. It is not clear if one can really neglect the terms linear in spin deviations when considering magneto-optical coupling, especially for the case of “spin-dominates” modes? Second, it seems that writing the same expressions for the spin-related order parameter components would give the same result, apart from the sign. Thus, it is not clear, if there is there any significant difference between the process of laser-induced excitation of the magnons in media with quenched and with unquenched angular momentum, apart from the magnon amplitudes?

6. An important statement made by the authors is that laser-induced excitation of magnons is highly-efficient in magnetic media with unquenched orbital momentum (page 6). This statement is not well substantiated. In experiments the measured value is $\Delta f(t)$ defined as $(I_1 - I_2)/(I_1 + I_2)$, where I_i is the intensity on the i-photodiode. This value, thus, is a measure of the changes of the dielectric permittivity caused by the pump-induced coherent magnons. The authors compare the amplitude of the oscillations Δf excited in CoO in the longitudinal geometry to those reported previously for NiO. However, this makes sense only if the deviations of the order parameter (e.g. L) or magnetization of sublattices from their equilibrium orientations are compared. Comparing Δf in two media, as it is introduced in the manuscript, does not provide directly information on the efficiency of excitation in two materials.

7. It would be useful to specify in the text the values describing the spin- and orbital-dominated modes. I think that the values of the ratios (S13,S15) should be listed in the main text, in order to make the manuscript accessible for a reader.

8. Fig. 1a. It is not clear what the direction of the x-axis is. It should be close to [110], is in't it?

Furthermore, although the manuscript is, in general, clearly written, some improvements are to be made:

1. The expression “light acting on magnetism” is a kind of jargon. In fact, the action of light on magnetic media, or on spin system, magnetic system etc. are being considered here.
2. It is not clear what the authors mean by “interactions mediated by magnetism....” (page 3). Exchange interaction is the origin of magnetic ordering, and not vice versa.
3. “Magnetism dependence of the... permittivity” (page 5) should be replaced by e.g. “dependence of permittivity on magnetic order parameter”.
4. “.. its sign changes from $\theta=45^\circ$” (page 5) should be replaced by e.g. “its sign changes when the pump polarization angle θ is changed from 45°”.
5. “...quadratic-to-magnetism terms....” (page 6) should be replaced by e.g. “terms quadratic in magnetic order parameter....”.

Reviewer #2 (Remarks to the Author):

The authors investigate the optical creation of magnon excitations in CoO using time-resolved fs pump-probe spectroscopy. They use the dependence of the optomagnetic coupling on the (incident) polarization angle to identify the different magnon modes and to ascribe to them their excitation energies.

CoO is an interesting candidate material for all-optical control of magnetism, because the magnetic moments are composed of both, spin and orbital angular momenta, due to strong spin-orbit coupling, and because the moments are not quenched in this material. This has two positive consequences for optical

control: (1) light couples strongly to the orbital degrees of freedom (because of electric dipoles) and, therefore, to the magnon excitations, and (2) the magnons have a relatively high energy reaching up to several THz, which is attractive for optical control.

These advantages of CoO over other transition metal oxides are well motivated in the paper. However, the presentation of the results and analysis appears rather poor. Several questions remain open or incompletely analyzed:

(1) In the the symmetry analysis of the magnon modes (bottom of page 6 in the pdf document) the authors use "irreducible represent combinations of the variables m_s and m_L ". I assume that they want to say that they use the irreducible representation of the *total* angular momentum $J=L+S$. Is this correct? If so, it should be clearly said.

In the same analysis, the authors introduce indexes 1 and 2 (bottom of page 6). From the context it appears that these indexes refer to a staggered magnetization. However, it is not even mentioned what the indexes refer to concretely.

(2) Why do the authors use an SU(3) theory to describe the collective modes? The spin/angular momentum dynamics is described by SU(2) whose characteristic is, in contrast to SU(3) that only transitions by one angular momentum unit are possible. The quench dynamics would rather require mixing of different irreducible SU(2) representations (different total angular momenta) than an SU(3).

Moreover, it is not explained what is the additional insight from this complete analysis for the experimental results. The modes had already been identified before by the polarization dependence.

(3) Fig 1 is rather unclear and the figure caption incomplete. In particular, in Fig 1 d, what are the different, vectors shown? What do the cones refer to? Without defining them, the drawings are rather meaningless, and the reader does not get insight from the figure which magnon modes are actually meant to describe.

Finally, a main result of the present experiments should be that the all-optical magnon mode excitation is strong compared to other materials, because of the strong coupling to the orbital momenta (see above). That is, the output-to-input signal ratio should be large. This would be especially relevant for applications the paper is alluding to. However, no analysis of this aspect is mentioned.

To conclude, the authors are able to identify specific magnon modes in CoO. This is a nice analysis. However, I believe that identifying modes alone, without, for instance, demonstrating unusually strong optomagnetic coupling, does not merit publication in Nature Communications. In addition, the presentation should be significantly improved.

Reviewer #3 (Remarks to the Author):

In this manuscript (NCOMMS-16-25539-T), Satoh and coauthors demonstrated ultrafast optical excitation of antiferromagnetic magnons in CoO up to 9 THz via inversed Cotton-Mouton effect (ICME) and inversed Faraday effect (IFE). The unquenched orbital angular momentum in CoO holds the key in the magnon dynamics due to the unusual symmetry of CoO, which results in much stronger magneto-optical effects than other transition metal oxide, e.g. NiO. Theoretical analysis on both magnon dynamics and magneto-optical effects are detailed in the supplement information to help assign the observed magnon modes.

The magnon properties of CoO has been studied both experimentally and theoretically since the 60s and magnon modes reported in this manuscript have been observed via Raman spectroscopy (Ref. [14]). But it is the first time that the high frequency (up to 9 THz) AFM magnon is coherently launched by ultrafast optical pulses and the subsequent magnon dynamics is studied systematically in the time domain. Although the experimental technique is routine, the selection of the sample, unique angle of the experiments and accompanying theory make the work novel and interesting. In general, I found the experimental observation are systematic and solid, and the theory part is also informative. However, I found the main claim that the unquenched orbital angular momentum gives rise to efficient excitation of magnon modes are not fully convincing, which may be related to the problematic logical flow and presentation. I found the main text is too brief with many essential components buried in the supplemental information (SI). I suggest a revision to improving the presentation and logic flow. The following comments/questions may help improve the presentation.

Comments / questions:

1. One of the most interesting claims that the unquenched orbital angular momentum in CoO strongly affects the excitation efficiency and the magnon dynamics has not adequately discussed throughout the main text, at least, not clearly presented for the general readers. First, the mechanism of unquenching and its relation to the structure has not been clearly shown in the main text. I would suggest moving some of the supplemental materials to the main text to make the connection between the reduced symmetry of the lattice structure and unquenched angular momentum clear. Second, both ICME and IFE are a general magneto-optical effect, not unique to

CoO. The “non-standard” ICME discussion in the main text may imply but does not directly support the unquenched orbital momentum. To me, one order of magnitude higher excitation efficiency of magnon in CoO comparing with that in NiO may be the only evidence that well-support the direct excitation of angular momentum in the unquenched CoO. Hope the revised manuscript can clearly list the evidence and provide detailed discussions to substantiate this main claim.

2. The assignment of magnon modes was discussed in the SI but not fully convincing.

(1) Two terms (“ $1-\cos(2\theta)$ ” and “ $\sin^2\theta \sin\psi$ ”) are presented in the phenomenological analysis in SI_3 Eq.S31. To assign 4.4THz to Γ_{2S} mode, it is implied that the first term should be dominant. But I am not sure why the domination of the first term is justified. Please explain.

(2) 6.6 THz mode is assigned to the Γ_{1S} because it has neither $\cos^2\theta$ nor $(1-\cos^2\theta)$ dependence. Is that possible the θ dependence in Fig.2c is simply offset by a θ -independent thermal contribution, as the author mentioned in the main text “Such helicity - independent excitations had been attributed to a thermal effect”. In addition, why was this 6.6THz not observed in the longitudinal geometry?

(3) 8.9 THz mode is assigned to degenerate Γ_1 and Γ_2 modes in which orbital contribution dominates. To me, 8.9 THz mode show very clear $\cos^2\theta$ dependence in transverse geometry, while it is too weak to draw a conclusion in longitudinal geometry. In addition, does the author have an explanation for why the low-freq spin dominated modes are non-degenerate in energy and high-freq orbit dominated modes are degenerate?

3. In Fig.1b, θ and ϕ label need to be swapped. According to the main text, the θ is associated with pump beam polarization plane, while ϕ is for the probe beam, but it shows the opposite in the current Fig.1b.

4. In the Fig.1 caption, it is better to add definitions for m_s , m_l , n_s and n_l , since they are explained in a rather later part of the main text.

5. In Fig.2b, it will be informative to show the magnon spectra at other θ and helicity, not just in 94 degrees.

Re: NCOMMS-16-25539-T

Excitation of coupled spin–orbit dynamics in cobalt oxide by femtosecond laser pulses by Takuya Satoh *et al.*

Dear Dr Gevaux,

First of all, we appreciate the time that the Reviewers devoted to assessing our manuscript. It was a great pleasure to find the three Reviewers referring to our study to be a “**novel result and of interest**”, “**well motivated**”, and “**novel and interesting**”. We are most grateful to all of the Reviewers for their comments and valuable suggestions, which have helped us to improve the manuscript.

We include our point-by-point responses to the Reviewers comments and queries as follows.

Responses to Reviewers' comments

Reviewer #1 (1):

The authors report on experimental study of laser-induced coherent dynamics of a magnetic order parameter in an antiferromagnet CoO. The feature of this compound is the unquenched orbital momentum, which is in contrast to iron oxides, extensively studied by femtosecond optical pump-probe methods in the recent years. The main finding reported in the manuscript is that four magnon modes are excited via inverse Faraday and Cotton-Mouton effects. The symmetries of the modes are determined from the pump-polarization dependences. **By performing comprehensive analysis of the magnon frequencies the authors determine contributions from orbital and spin momenta to these modes.**

The experimental demonstration of coherent dynamics of magnetic order parameter in the compound with unquenched orbital momentum, excited via ultrafast inverse magneto-optical effects is **the novel result and of interest to the community working in the field of femtomagnetism.**

Reviewer #1 (2):

However, the extended theoretical analysis mainly concerns the frequencies of the modes, and the experimental data on excitation of these modes are required only to get the symmetry of the modes. In fact, Raman scattering experiments could yield the same information (Suppl. information, Section 6). Therefore, the manuscript in general does not provide insight into yet unexplored features of coupling of femtosecond pulses to compounds with unquenched orbital momentum. Thus, although the experimental observations and the theoretical analysis are of interest to the particular community of researchers, I cannot recommend publication of the manuscript in the present form in Nature Communications.

Response:

We fully agree with Reviewer #1 that spontaneous magnon Raman scattering gives information about magnon modes with frequencies and their symmetries. Nevertheless, we stress that we have not only investigated magnon modes, but demonstrated, for the

first time, the possibility of *coherent* excitation of these modes by *femtosecond laser pulses*. Thus, our finding indeed provides insight into the unexplored coupling of femtosecond pulses to compounds with unquenched orbital angular momentum (OAM); specifically, our article investigates this aspect with the CoO as a model system. The specific role of angular momentum is found in the larger coupling strength of light with OAM compared with spin. We show that the signal for CoO having an unquenched OAM is much higher than for compounds with quenched OAM, e.g., for NiO.

Please also refer to the positive evaluation by Reviewer #3 (1) of the novelty of the ultrafast and efficient excitation of high-frequency magnons.

Following Reviewer #1's comments, we added the above discussion to the revised manuscript as follows.

Lines 49–50: *Further, the coupling...*

Lines 51–54: *In the following, ...*

Lines 166–169: *The amplitude of oscillations...*

Lines 227–228: *To summarize, we demonstrated that...*

Supplementary Note 4 (Pages 13–14).

Reviewer #1 (3):

Below I list the specific issues which the authors should address before the manuscript can be considered for publication.

1. Supplementary information. Parts 1-2. Presented theoretical analysis seems to be the key element of the manuscript. Therefore, the validity of the model, considered here, should be carefully substantiated. The authors introduce the Hamiltonian (S1), which includes distinct terms describing anisotropy for the orbital moments and for spins. It is generally accepted that the magneto-crystalline anisotropy originates from the spin-orbital coupling. Thus, it is not clear, what is the source of anisotropy for spins, if not the coupling to orbital moments. Could the authors give a brief explanation of the physics behind different terms in Eq. (S1). It is necessary, since the manuscript is meant for a broad research community.

Response:

We concur with Reviewer #1 and have added clarifying remarks on this aspect. Indeed, for Co^{2+} ions in CoO, partial quenching of the OAM L takes place, which reduces its $L=3$ of the free state to $L=1$ in the cubic crystal field. This feature gives rise both to the appearance of “almost-free” effective OAM $L=1$ and to a significant contribution to the spin anisotropy from the partially quenched part.

We have inserted a short description of this important issue in the manuscript:

Lines 34–46: *For CoO,...*

Supplementary Note 1 (Page 5): *The contribution of spin-orbit interaction to the spin anisotropy...*

Reviewer #1 (4):

2. The magnon mode of 8.9 THz in the LG geometry is extremely weak and its amplitude in the FFT spectrum (Fig. 3c) seems not to exceed the noise level. I suggest the authors

to present more experimental data, proving that the 8.9 THz peak in the FFT spectra is a reliable one (e.g. data for the probe polarization of 135 deg, at which the amplitude of the 8.9 THz oscillations seems to be the largest (Fig. S2b)). How important the fact that this mode is observed for the conclusions of the manuscript?

Response:

We accept the Reviewer #1's suggestion. The observation of the 8.9-THz mode was indispensable for our conclusion regarding the mode identification. Indeed, the magnon mode of 8.9 THz was clearly observed in the TG. In the LG, however, the signal is expected to be still too weak even for the probe polarization of 135°. To confirm the 8.9-THz mode in the LG, we performed spontaneous Raman scattering measurements; see Supplementary Figure 2b (Page 2). For a symmetry argument, spontaneous Raman scattering is sufficient. The result clearly supports our conclusion.

Reviewer #1 (5):

3. Discussion of the results presented in Fig. 2c. The authors write that the polarization dependences of the amplitudes of the modes 6.6 and 8.9 THz are proportional to $(\cos(2\theta)+\text{const})$ and $\cos(2\theta)$, which contradicts the data in Fig. 2c. In fact, from this Figure one can see that the amplitudes of all three modes observed in the given experimental geometry can be described by a unified expression $(C-A\cos(2\theta))$, where A and C take specific values for each mode.

Response:

As pointed out by the Reviewer #1, indeed, all the experimental results can be expressed in a unified form by $C-A\cos(2\theta)$. Here the important point for mode identification is that the theory gives only $C=0$ or $C=A$, and not arbitrary values.

Following Reviewer #1's comment, we have pointed this out in the revised manuscript. Line 109–110: *and $(\cos 2\theta + \text{const.})$ for the 6.6-THz mode,...*

Reviewer #1 (6):

4. It is not clear for me, why the 4.4-THz mode excited in the transversal geometry by circularly-polarized pump pulses is insensitive to the pump helicity. From Eq. (S1-S32) it follows that there should be a clear sign-change, if one takes into account strong ellipticity of the $\Gamma_2(S)$ mode.

Response:

We concur with Reviewer #1's comment. The observed 4.4-THz mode in the TG is attributed to the ICME from the term G_4 in Supplementary Equation (31), which can also be excited by circular-polarized pump pulses with no helicity dependence. In principle, this mode can be excited by the IFE because of the term K_2 in Supplementary Equation (32) as well as the ICME from term G_4 in Supplementary Equation (31). The reason why this mode was not excited by the IFE is the following. The 4.4-THz mode is a $\Gamma_2(S)$ mode, where the trajectory of the magnetization vector is an ellipse elongated along the y axis. Excitation by the IFE (corresponding to the K_2 term in Supplementary Equation (32)) initializes as a kick on the magnetization along the y direction whereas that by the ICME (corresponding to the G_4 term in Supplementary Equation (31)) is along the x direction,

which corresponds to the short axis of the ellipse. Therefore, the ICME dominates the excitation efficiency, and therefore the helicity-dependent excitation via the IFE was not observed.

Following Reviewer #1's comment, we have transferred a relevant passage from the Supplementary Information to the main text. Accordingly, the structure of the main text has been changed.

Lines 70–80: *We introduce convenient combinations...*

Lines 110–112: *Surprisingly, for a circularly polarized...*

Lines 113–140: *To explain this complicated picture,...*

Lines 219–226: *Here we discuss the reason why...*

Reviewer #1 (7):

5. When analyzing the coupling of light to the excited modes (Suppl. Material, section 3) the authors consider only the terms linear on dynamic components of orbital momentum. It is not clear if one can really neglect the terms linear in spin deviations when considering magneto-optical coupling, especially for the case of “spin-dominates” modes? Second, it seems that writing the same expressions for the spin-related order parameter components would give the same result, apart from the sign. Thus, it is not clear, if there is there any significant difference between the process of laser-induced excitation of the magnons in media with quenched and with unquenched angular momentum, apart from the magnon amplitudes?

Response:

First of all, we point out that in the expression of n_l in Supplementary Equation (17), the subscript “ l ” is not related to the OAM, but a lower index for a tensor “ $ijkl$ ”. The symmetric properties of the oscillations of the corresponding spin and orbital vectors are indeed the same, as well as the symmetric properties of the tensors k_{ijk} and g_{ijkl} . Although both the orbital and spin angular momenta contribute to the magnetism in CoO, the magneto-optical effect is dominated by the OAM because the optical selection rule for the electric-dipole interaction only allows changes in OAM. A change in spin can be only allowed via spin-orbit interactions. In many other systems, however, the OAM is quenched and cannot contribute to the magnetic properties of media, and thus magneto-optical effect is governed by the indirect coupling between spin and light.

To simplify the expressions, we omit the index “ L ” from vectors m_L, n_L .

We have added a remark addressing this point to the text.

Lines 119–120: *Note that the subscript...*

Lines 123–132: *The symmetric properties of the oscillations...*

Reviewer #1 (8):

6. An important statement made by the authors is that laser-induced excitation of magnons is highly-efficient in magnetic media with unquenched orbital momentum (page 6). This statement is not well substantiated. In experiments the measured value is $\Delta f(t)$ defined as $(I_1 - I_2)/(I_1 + I_2)$, where I_i is the intensity on the i -photodiode. This value, thus, is a measure of the changes of the dielectric permittivity caused by the pump-induced coherent magnons. The authors compare the amplitude of the oscillations $\Delta f(t)$

f excited in CoO in the longitudinal geometry to those reported previously for NiO. However, this makes sense only if the deviations of the order parameter (e.g. L) or magnetization of sublattices from their equilibrium orientations are compared. Comparing Δf in two media, as it is introduced in the manuscript, does not provide directly information on the efficiency of excitation in two materials.

Response:

We concur with the Reviewer #1's comment as it suggests a significant improvement highlighting the novelty of the study. The yield of the impulsive stimulated Raman scattering under non-resonance conditions and hence the magnitudes of the pump-probe signals for a given pump and probe fluence are determined by the Raman tensor, which depends on various parameters such as magnon frequency, as well as optical frequencies and polarizations of the pump and probe light (L. Dhar *et al.*, Chem. Rev. 94, 157–193 (1994)). Therefore, it is not realistic to determine the Raman tensor elements for all possible magnonic and optical parameters for a direct comparison of the cases of CoO and NiO. Alternatively, in the revised manuscript, we compare the output-to-input signal ratios with an equivalent pump fluence, as suggested by the Reviewer #2 (see the comment Reviewer #2 (6) and our answer on it for details). Furthermore, we notice that the magnon excitation efficiency depends on the temporal durations of the pump and probe pulses and the magnon frequency, which is implemented as pump-probe spectral weights (for definition see Supplementary Equation (37) in Supplementary Note 4). These normalizations are sufficient for an order-of-magnitude estimation of the coupling strength for comparing the spin-orbit coupled system (CoO) and the spin-mediated system (NiO). As a result, we conclude that the output-to-input signal ratio is about two orders-of-magnitude larger for CoO.

We also note that unlike the spontaneous Raman scattering, the Bose-Einstein factor $(n+1)$, see e.g., Eq. (6.66) in the textbook by Heyes and Loudon, "Scattering of light by crystals," is not present for pump-pulse-induced coherent magnons.

We have added remarks covering the above discussion to the revised manuscript.

Lines 166–169: *The amplitude of oscillations...*

Supplementary Note 4 (Pages 13–14).

Reviewer #1 (9):

7. It would be useful to specify in the text the values describing the spin- and orbital-dominated modes. I think that the values of the ratios (S13,S15) should be listed in the main text, in order to make the manuscript accessible for a reader.

Response:

We calculated the ellipticities of spin and orbital precessions. Ratios of the amplitudes for variables m_S , m_L , n_S , and n_L are given in the caption of Fig. 1b (Lines 358–362).

Reviewer #1 (10):

8. Fig. 1a. It is not clear what the direction of the x-axis is. It should be close to $[110]$, is in't it?

Response:

The directions for the three axes are x : [772], y : [-110], z : [-1-17].
We have added them to Fig. 1a.

Reviewer #1 (11):

Furthermore, although the manuscript is, in general, clearly written, some improvements are to be made:

1. The expression “light acting on magnetism” is a kind of jargon. In fact, the action of light on magnetic media, or on spin system, magnetic system etc. are being considered here.

Response:

We agree with Reviewer #1 and have changed the expression.
Lines 23–24: *light acting on magnetic systems*,

Reviewer #1 (12):

2. It is not clear what the authors mean by “interactions mediated by magnetism....” (page 3). Exchange interaction is the origin of magnetic ordering, and not vice versa.

Response:

We agree with Reviewer #1 and have completely rewritten the relevant paragraph.
Lines 34–46: *For CoO*,...

Reviewer #1 (13):

3. “Magnetism dependence of the... permittivity” (page 5) should be replaced by e.g. “dependence of permittivity on magnetic order parameter”.

Response:

We agree with Reviewer #1 and have changed this expression.
Lines 114–115: *dependence of the permittivity on the magnetic order parameter*.

Reviewer #1 (14):

4. “.. its sign changes from $\theta=45\dots$ ” (page 5) should be replaced by e.g. “its sign changes when the pump polarization angle θ is changed from $45\dots$ ”.

Response:

We agree with Reviewer #1 and have changed this expression.
Lines 146–147: *its sign changes when the pump azimuth angle θ is changed from $45\dots$*

Reviewer #1 (15):

5. “...quadratic-to-magnetism terms....” (page 6) should be replaced by e.g. “terms quadratic in magnetic order parameter....”.

Response:

We agree with Reviewer #1 and have changed this expression.
Line 152: *other G-related terms*,...

Reviewer #2 (1):

The authors investigate the optical creation of magnon excitations in CoO using time-resolved fs pump-probe spectroscopy. They use the dependence of the optomagnetic coupling on the (incident) polarization angle to identify the different magnon modes and to ascribe to them their excitation energies.

CoO is an interesting candidate material for all-optical control of magnetism, because the magnetic moments are composed of both, spin and orbital angular momenta, due to strong spin-orbit coupling, and because the moments are not quenched in this material. This has two positive consequences for optical control: (1) light couples strongly to the orbital degrees of freedom (because of electric dipoles) and, therefore, to the magnon excitations, and (2) the magnons have a relatively high energy reaching up to several THz, which is attractive for optical control. **These advantages of CoO over other transition metal oxides are well motivated in the paper.**

Reviewer #2 (2):

However, the presentation of the results and analysis appears rather poor. Several questions remain open or incompletely analyzed:

(1) In the the symmetry analysis of the magnon modes (bottom of page 6 in the pdf document) the authors use "irreducible represent combinations of the variables m_s and m_L ". I assume that they want to say that they use the irreducible representation of the *total* angular momentum $J=L+S$. Is this correct? If so, it should be clearly said.

In the same analysis, the authors introduce indexes 1 and 2 (bottom of page 6). From the context it appears that these indexes refer to a staggered magnetization. However, it is not even mentioned what the indexes refer to concretely.

Response:

To avoid the misunderstanding with this terminology, we have changed "irreducible combinations of the variables" to "convenient combinations of the variables."

Line 70: *convenient combinations of the variables.*

The labels 1 and 2 refer to the spin and orbital momenta for 1-st and 2-nd sublattices of the antiferromagnetic CoO.

We have insert a clarifying remark in the revised manuscript.

Line 60: *(labeled with 1 and 2).*

Reviewer #2 (3):

(2) Why do the authors use an SU(3) theory to describe the collective modes? The spin/angular momentum dynamics is described by SU(2) whose characteristic is, in contrast to SU(3) that only transitions by one angular momentum unit are possible. The quench dynamics would rather require mixing of different irreducible SU(2) representations (different total angular momenta) than an SU(3).

Response:

In response to this comment we have added to Supplementary Note 1 a more detailed explanation as to why the standard spin-wave theory, in accounting for transitions with changes of angular momentum by one, is not valid here. This kind of theory can be

classified as SU(2) coherent-state theory; also it is known as spin coherent-state theory; see Refs. on the reviews/books of Perelomov and Fradkin [7,8] of the new Supplementary Note. This theory operates with only quantum expectation values of spin components and it is equivalent to the Landau–Lifshitz equation. Contrary to this, SU(3) coherent-state theory accounts for all the possible transitions for the angular momentum-one state, and works for both dipolar variables (mean values of the angular momentum components) and quadrupolar spin variables (mean values of operators, bilinear on the angular momentum components). Note that the terminology [SU(3) coherent states or SU(2) coherent states] is not connected directly with the fact that the angular momentum generator of the group of rotations SO(3)~SU(2); the fact is the full set of dipolar and quadrupolar operators span the algebra of SU(3) group. This theory allow for a description of the full dynamics of a angular momentum-one system that is employed in our theory for effective OAM $L=1$.

We have added a discussion to Supplementary Note 1 to make the problem clearer (Page 6).

Reviewer #2 (4):

Moreover, it is not explained what is the additional insight from this complete analysis for the experimental results. The modes had already been identified before by the polarization dependence.

Response:

The symmetry analysis provides the only description of the polarization dependence of the input/output signal. The theory gives the analytical dependences of all the characteristics of the magnon modes on the parameters of the Hamiltonian. This result allows us **not only** to find the aforementioned parameters by fitting the observed values of frequencies, **but also** to calculate the ratios of the amplitudes of the orbital /spin momenta and antiferromagnetic vector/magnetization for a given mode. This leads to an important insight into the properties of the modes and their interaction with light.

We have added this description to the caption of Fig. 1b (Lines 358–362).

Reviewer #2 (5):

(3) Fig 1 is rather unclear and the figure caption incomplete. In particular, in Fig 1 d, what are the different, vectors shown? What do the cones refer to? Without defining them, the drawings are rather meaningless, and the reader does not get insight from the figure which magnon modes are actually meant to describe.

Response:

As suggested by Reviewer #2, the figure caption has been fully revised.
Lines 350–358: *Crystallographic and magnetic structures of CoO...*

Reviewer #2 (6):

Finally, a main result of the present experiments should be that the all-optical magnon mode excitation is strong compared to other materials, because of the strong coupling to

the orbital momenta (see above). That is, the output-to-input signal ratio should be large. This would be especially relevant for applications the paper is alluding to. However, no analysis of this aspect is mentioned.

Response:

We deeply appreciate Reviewer #2's proposal on a quantitative analysis for comparing the present experiment with the previous studies. In this comparison, output-to-input ratios are a realistic way to estimate the efficiency of the optical read-out of optically written information stored by magnons.

We have extracted these ratios from the experiment data, and compared the values with other experiments. For comparing different experiments, we normalized the output-to-input ratio by the pump fluence, which is proportional to the excitation efficiency. Furthermore, we notice that the magnon excitation efficiency depends on the temporal durations of the pump and probe pulses and the magnon frequency, which is implemented as pump-probe spectral weights (for a definition, see Supplementary Equation (37) in Supplementary Note 4). These normalizations are sufficient for an order-of-magnitude estimation of the coupling strength for comparing the spin-orbit coupled system (CoO) and the spin-mediated system (NiO). As a result, we conclude that the output-to-input signal ratio is about two orders-of-magnitude larger for CoO.

We have inserted remarks outlined in the above in the revised manuscript.

Lines 166–169: *The amplitude of oscillations...*

Supplementary Note 4 (Pages 13–14).

Reviewer #2 (7):

To conclude, the authors are able to identify specific magnon modes in CoO. This is a nice analysis. However, I believe that identifying modes alone, without, for instance, demonstrating unusually strong optomagnetic coupling, does not merit publication in Nature Communications. In addition, the presentation should be significantly improved.

Response:

We emphasize that we have not only identified modes but also obtained strong output-to-input ratios from the extended analysis. We believe that the revised version of the manuscript is significantly improved. Please also see the positive comment of Reviewer #3 (1).

Reviewer #3 (1):

In this manuscript (NCOMMS-16-25539-T), Satoh and coauthors demonstrated ultrafast optical excitation of antiferromagnetic magnons in CoO up to 9 THz via inversed Cotton-Mouton effect (ICME) and inversed Faraday effect (IFE). The unquenched orbital angular momentum in CoO holds the key in the magnon dynamics due to the unusual symmetry of CoO, which results in much stronger magneto-optical effects than other transition metal oxide, e.g. NiO. Theoretical analysis on both magnon dynamics and magneto-optical effects are detailed in the supplement information to help assign the observed magnon modes.

The magnon properties of CoO has been studied both experimentally and theoretically since the 60s and magnon modes reported in this manuscript have been observed via

Raman spectroscopy (Ref. [14]). But it is the first time that the high frequency (up to 9 THz) AFM magnon is coherently launched by ultrafast optical pulses and the subsequent magnon dynamics is studied systematically in the time domain. Although the experimental technique is routine, the selection of the sample, unique angle of the experiments and accompanying theory make the work novel and interesting. In general, I found the experimental observation are systematic and solid, and the theory part is also informative.

Response:

We are glad that the reviewer finds our study “novel” and “interesting,” and agrees that this is the first demonstration of *coherent* driving of the high frequency AFM modes by *femtosecond laser pulses*, systematically studied in *the time domain*, employing CoO as a model system. This is indeed the central value of this study, providing insights into the unexplored coupling of femtosecond pulses to compounds with unquenched OAM. Emphatically, the reviewer has found that in general the experimental observation “systematic” and “solid” as well as the theoretical analysis “detailed” and “informative.”

Reviewer #3 (2):

However, I found the main claim that the unquenched orbital angular momentum gives rise to efficient excitation of magnon modes are not fully convincing, which may be related to the problematic logical flow and presentation. I found the main text is too brief with many essential components buried in the supplemental information (SI). I suggest a revision to improving the presentation and logic flow.

Response:

We concur that this suggestion does improve the presentation in highlighting the efficient excitation of magnon modes, which is indeed the main consequence of the presence of the unquenched OAM in CoO. We agree that the original version of the manuscript lacked a sufficient and quantitative description of the efficiency. As discussed later (Reviewer #3 (3)), we show in the revised manuscript that the efficiency for CoO having unquenched OAM is two-orders-of-magnitude higher than for NiO, a similar but quenched system.

We discuss the reasons why the systems with unquenched OAM can exhibit efficient light-magnetism coupling in the revised manuscript:

Lines 6–12: *Visible or near-infrared light...*

Lines 125–131: *Although both the orbital and spin angular momenta...*

Supplementary Note 3 (Page 12): *Although both the orbital and spin angular momenta...*

Following the reviewer’s suggestion, we also transferred a relevant passage from Supplementary Note to the main text. Accordingly, the structure of the main text has been changed.

Lines 70–80: We introduce convenient combinations

Lines 113–140: *To explain this complicated picture,...*

Furthermore, we add Supplementary Note 4 to show details of the method for comparing the efficiencies of the magnon excitation in CoO and NiO.

In complying with the Reviewer #3's suggestion, we have made a major revision of the main text and the Supplementary Note throughout to improve the logical flow. We believe that the revised manuscript now contains a convincing argument to support our main claims.

Reviewer #3 (3):

The following comments/questions may help improve the presentation.

Comments / questions:

1. One of the most interesting claims that the unquenched orbital angular momentum in CoO strongly affects the excitation efficiency and the magnon dynamics has not adequately discussed throughout the main text, at least, not clearly presented for the general readers.

Response:

We have followed the advice of the reviewer to improve the presentation. This first suggestion to highlight the excitation efficiency is essentially the same as comments by Reviewer #1 (8) and Reviewer #2 (6). We compared the output-to-input signal ratio with an equivalent pump fluence. We further took into account that the magnon excitation efficiency depends on the time durations of the pump and probe pulses and the magnon frequency, which is implemented as the pump-probe spectral weights (for definition see Supplementary Equation (37) in Supplementary Note 4). An order-of-magnitude estimation suffices for the spin-orbit coupling dynamics. As a result, we conclude that the output-to-input signal ratio of CoO is two-orders-of-magnitude larger than that of NiO.

We have revised the manuscript accordingly.

Lines 166–169: *The amplitude of oscillations...*

Supplementary Note 4 (Pages 13–14).

Reviewer #3 (4):

First, the mechanism of unquenching and its relation to the structure has not been clearly shown in the main text. I would suggest moving some of the supplemental materials to the main text to make the connection between the reduced symmetry of the lattice structure and unquenched angular momentum clear.

Response:

The discussion of the mechanism of unquenching and its physical sequences has been added to the main text as well as the Supplementary Note.

Lines 34–46 : *For CoO,...*

Supplementary Note 1 (Page 4): *The free Co²⁺ ion has the electronic structure...*

Supplementary Note 1 (Page 4): *It also has been mentioned that...*

Reviewer #3 (5):

Second, both ICME and IFE are a general magneto-optical effect, not unique to CoO. The “non-standard” ICME discussion in the main text may imply but does not directly support the unquenched orbital momentum. To me, one order of magnitude higher excitation efficiency of magnon in CoO comparing with that in NiO may be the only evidence that well-support the direct excitation of angular momentum in the unquenched CoO. Hope

the revised manuscript can clearly list the evidence and provide detailed discussions to substantiate this main claim.

Response:

As with comments by Reviewer #1 (8), Reviewer #2 (6), and Reviewer #3 (3), appreciatively, we have revised the manuscript:

Lines 49–50: *Further, the coupling...*

Lines 166–169: *The amplitude of oscillations...*

Supplementary Note 4 (Pages 13–14).

Concerning the first part of the above comment, we agree that it is better to say “non-standard manifestation of ICME” instead of “non-standard ICME” in that the former corresponds rather to the low symmetry of CoO than to its orbital dynamics.

We have made changes to the text reflecting this perspective.

Lines 162–164: *This helicity-independent excitation of the magnon...*

Reviewer #3 (6):

2. The assignment of magnon modes was discussed in the SI but not fully convincing.

(1) Two terms (“ $1-\cos^2(\theta)$ ” and “ $\sin^2(\theta) \sin^2(\psi)$ ”) are presented in the phenomenological analysis in SI_3 Eq.S31. To assign 4.4THz to $\Gamma_{2,S}$ mode, it is implied that the first term should be dominant. But I am not sure why the domination of the first term is justified. Please explain.

Response:

This is solely an experimental result. The symmetry argument based on group theory cannot predict which term is dominant. If the term $G_4(1-\cos^2(\theta))$ is dominant in Supplementary Equation (31), it should lead to a $(1-\cos^2(\theta))$ dependence for linear polarization and to a contribution for circular polarization. Indeed, our experiment results in Fig. 2c and d show just this feature for both polarizations. Dependences characteristic of the term with $\sin^2(\theta) \sin^2(\psi)$ were not seen. In contrast, if $G_2(1-\cos^2(\theta))$ or $G_3(1-\cos^2(\theta))$ is dominant in Supplementary Equation (29) in the TG, then in the LG $G_2(1+\sin^2(\theta) \cos^2(\psi))$ or $G_3(1-\sin^2(\theta) \cos^2(\psi))$ would be present but contradicts the experimental result ($\cos^2(\theta)$). Therefore, we conclude that G_4 is the dominant factor and the 4.4-THz mode is Γ_2 .

Reviewer #3 (7):

(2) 6.6 THz mode is assigned to the $\Gamma_{1,S}$ because it has neither $\cos^2(\theta)$ nor $(1-\cos^2(\theta))$ dependence. Is that possible the θ dependence in Fig.2c is simply offset by a θ -independent thermal contribution, as the author mentioned in the main text “Such helicity -independent excitations had been attributed to a thermal effect”. In addition, why was this 6.6THz not observed in the longitudinal geometry?

Response:

Indeed, the thermal effect cannot be fully excluded. The reason why the 6.6-THz mode was not observed in the LG is not clear. To confirm that the 6.6-THz mode is Γ_1 , we performed spontaneous Raman scattering in the LG. The polarization dependence

clearly points to Gamma 1. A statement addressing this point has been added to Supplementary Figure 2b (Page 2).

Reviewer #3 (8):

(3) 8.9 THz mode is assigned to degenerate Γ_1 and Γ_2 modes in which orbital contribution dominates. To me, 8.9 THz mode show very clear $\cos^2\theta$ dependence in transverse geometry, while it is too weak to draw a conclusion in longitudinal geometry. In addition, does the author have an explanation for why the low-freq spin dominated modes are non-degenerate in energy and high-freq orbit dominated modes are degenerate?

Response:

Here we suggest that this is just an experimental result. Our model allows for the description of this result by a corresponding choice of constants, and the extraction of the constants from the experimental data, but we cannot calculate these constants.

Because the 6.6-THz mode is assigned to Γ_1 , $G_1+G_2+G_3$ in Supplementary Equation (29) is non-zero. The $\cos^2\theta$ dependence of the 8.9-THz mode in the TG can be understood if $G_1+G_2+G_3=G_1-G_2-G_3=G_4$ and the 8.9-THz mode is mixture of Γ_1 (Supplementary Equation (29)) and Γ_2 (Supplementary Equation (31)).

Indeed, the spontaneous Raman scatterings in the TG and LG in Supplementary Figure 2a and b clearly show that the peak corresponding to the 8.9-THz mode is a mixture of Γ_1 and Γ_2 .

Reviewer #3 (9):

3. In Fig.1b, θ and ϕ label need to be swapped. According to the main text, the θ is associated with pump beam polarization plane, while ϕ is for the probe beam, but it shows the opposite in the current Fig.1b.

Response:

As pointed out by Reviewer #3, corrections have been made in Fig. 1 b and c.

Lines 363–365: *θ and ϕ denote the azimuthal angles of...*

Reviewer #3 (10):

4. In the Fig.1 caption, it is better to add definitions for ms, ml, ns and nL, since they are explained in a rather later part of the main text.

Response:

We concur with this suggestion, and the caption has be revised.

Lines 355–358: *Spin (S_1, S_2) and orbital (L_1, L_2) angular...*

Reviewer #3 (11):

5. In Fig.2b, it will be informative to show the magnon spectra at other θ and helicity, not just in 94 degrees.

Response:

We found that the magnon spectra at other pump polarization do not provide any useful information. Therefore, we have left it as is.

Reviewer #1 (Remarks to the Author):

Sato et al. have revised the manuscript in order to address the comments of the referees. In particular, following the suggestion of the referees, the authors have clarified, how the efficiency of the magnons excitation was defined, and show that the comparison between the results in CoO and NiO is justified.

I can recommend the acceptance of the manuscript after the following issues are addressed.

In my opinion, the revised discussion of the amplitude of the signal in CoO, as compared to NiO is, indeed, interesting. However, it is mainly done in the supplementary information, while the discussion in the main text is brief and not concrete. I recommend to revise the abstract and the main text (page 8), in order to provide a more specific discussion of the efficiency. In particular, the authors should clearly state that it is the output-to-input ratio they estimate. Further, I would like to stress, that the efficiency the authors discuss, is not exactly the efficiency of the magnon mode excitation. It is the efficiency of the probe polarization modulation, given in [rad of the polarization rotation per mJ/cm²].

Another issue, which I recommend the authors to address, is the role of birefringence. Since there is static birefringence in the crystal, how does it affect the observed modulation of the probe polarization? Pump and probe pulses inside the crystal should experience the changes of the polarization due to birefringence, which could affect both, the excitation efficiency and the detection [J. Phys.: Condens. Matter 29 164004; J. Appl. Phys. 101, 053912]. Was it taken into account?

Further, there are a few minor comments:

Main text:

Page 8. Something is wrong with the phrase "The temperature dependence of the spectral magnitude again confirms the origin of magnetism in CoO..".

Figs. 3. There is inconsistency between data in the panel (a), in the text, and in panel (c). While in the text the amplitude of the 4.4 THz mode is said to be of about 0.02, in the panel (c) it reaches ~30. The same concerns Fig. 2.

Supplementary information:

Fig. 2. The frequencies should be given in THz, as in the rest of the manuscript.

Reviewer #2 (Remarks to the Author):

The authors have substantially improved the presentation and essentially eliminated the concerns of my previous report.

They corrected the description of their magnetization and sublattice variables.

They clarified that they use an SU(3) theory (rather than SU(2)) in order to incorporate not only dipole but also quadrupole transitions. This seems to be needed here since the light field can induce $\Delta L = \pm 2$ (quadrupolar) transitions of the orbital part of the total angular momentum. This is an interesting aspect, now better explained in a new supplementary note.

They have extended the explanation of Fig. 1, although I am still somewhat doubtful about the readability of this figure.

Finally, and most importantly, the authors have performed an additional analysis of the output-to-input ratio, as suggested in my previous report. This analysis shows a substantially larger response of the CoO over other systems. I believe that this significantly enhances the importance of this paper and

makes it overall suitable for Nature Communications.

I have been asked to comment also on the authors' response to the other reviewers' concerns. Some of these concerns coincide or have large overlap with my own previous concerns, as also recognized by the authors. I think that the authors thoroughly answered these concerns as well.

Therefore I can now recommend the paper for publication.

Reviewer #3 (Remarks to the Author):

After reading the referee's reports and responses, the general remark from the previous round review is that the observation and experiments are potentially interesting but the original version of the manuscript has issues with presentation and insufficient discussion so that it was not acceptable for publication. In the revised manuscript, the authors have made important changes and improvement based on the suggestions from the reviewers. Most of the responses to my concerns are reasonable and satisfactory. However, there are still following questions that need to be addressed before I can recommend it for publication.

1. In the response to Reviewer #3 (11), I don't understand why the authors say "the magnon spectra at other pump polarization do not provide useful information". Wasn't the Fig.2c,d obtained by analyzing magnon spectra? Based on the time-domain traces shown in Fig.2a, their spectra should be quite different. For example, comparing the trace at 94 degrees, the oscillation period measured at 4 degrees are much smaller. It means the high-frequency component should be dominant at 4 degrees. The authors should compare these spectra and discuss the difference.
2. In supplementary figure 2a, authors assigned a peak at 200 cm⁻¹ to the 6.6 THz mode. However, there is another peak at 220 cm⁻¹, corresponding exactly to 6.6 THz mode. Why do the authors assign 220 cm⁻¹ peak to 6.6 THz mode? What is the origin of the 200 cm⁻¹ Raman peak?
3. In supplementary figure 2b, could author explain why the peak widths are much broader than the ones shown in Fig. 2b of the main text?

Re: NCOMMS-16-25539-T

Excitation of coupled spin–orbit dynamics in cobalt oxide by femtosecond laser pulses by Takuya Satoh *et al.*

Dear Dr Gevaux,

First of all, we appreciate the time that the reviewers devoted to assessing our manuscript. We are most grateful to all of the reviewers for their valuable comments and suggestions, which have helped us to improve the manuscript. Changes to the manuscript and Supplementary Note are marked in red. Two versions of the manuscript are provided, with and without the revision marks.

We include our point-by-point responses to the reviewers' comments as follows.

Responses to the reviewers' comments

Reviewer #1 (16):

Satoh et al. have revised the manuscript in order to address the comments of the referees. In particular, following the suggestion of the referees, the authors have clarified, how the efficiency of the magnons excitation was defined, and show that the comparison between the results in CoO and NiO is justified.

I can recommend the acceptance of the manuscript after the following issues are addressed.

Response:

We thank Reviewer #1 for their positive evaluation of our work.

Reviewer #1 (17):

In my opinion, the revised discussion of the amplitude of the signal in CoO, as compared to NiO is, indeed, interesting. However, it is mainly done in the supplementary information, while the discussion in the main text is brief and not concrete. I recommend to revise the abstract and the main text (page 8), in order to provide a more specific discussion of the efficiency. In particular, the authors should clearly state that it is the output-to-input ratio they estimate. Further, I would like to stress, that the efficiency the authors discuss, is not exactly the efficiency of the magnon mode excitation. It is the efficiency of the probe polarization modulation, given in [rad of the polarization rotation per mJ/cm²].

Response:

We agree with Reviewer #1 and have revised the main text as follows.

Lines 10–13: *We demonstrate excitations...*

Lines 140–142: *The output-to-input ratio...*

Reviewer #1 (18):

Another issue, which I recommend the authors to address, is the role of birefringence. Since there is static birefringence in the crystal, how does it affect the observed modulation of the probe polarization? Pump and probe pulses inside the crystal should

experience the changes of the polarization due to birefringence, which could affect both, the excitation efficiency and the detection [J. Phys.: Condens. Matter 29 164004; J. Appl. Phys. 101, 053912]. Was it taken into account?

Response:

We agree with Reviewer #1 that the role of birefringence should be addressed in the evaluation of the output-to-input ratio. In the two-color pump-probe experiment, the birefringence suppresses the output-to-input ratio depending on the pump and probe wavelengths and sample thickness. As we wrote in the Methods, the CoO crystal in the LG has a static birefringence of $\Delta n \sim 0.001$, which leads to a phase shift of $\sim \pi/10$ ($\ll 2\pi$) between the ordinary and extraordinary components of the pump/probe beams. Thus, one can assume that the pump beam maintains its polarization through the sample. Note that in the experiment with a NiO crystal in ref. 23, the authors chose a magnetic domain that was free from birefringence. Therefore, the presence of slight static birefringence in CoO does not influence the conclusion of the orders-of-magnitude comparison of the output-to-input ratios between the CoO (in the LG) and NiO experiments.

We added the above discussion to the Supplementary Note as follows.
Supplementary Note 4 (Page 14).
Supplementary References

Reviewer #1 (19):

Further, there are a few minor comments:

Main text:

Page 8. Something is wrong with the phrase “The temperature dependence of the spectral magnitude again confirms the origin of magnetism in CoO..”.

Response:

We agree with Reviewer #1 and have revised the main text as follows.
Lines 146–147: *The temperature dependence...*

Reviewer #1 (20):

Figs. 3. There is inconsistency between data in the panel (a), in the text, and in panel (c). While in the text the amplitude of the 4.4 THz mode is said to be of about 0.02, in the panel (c) it reaches ~ 30 . The same concerns Fig. 2.

Response:

We thank Reviewer #1 for pointing out this issue. We have changed Fig. 2b and c and Fig. 3b and c to correct this inconsistency.

Reviewer #1 (21):

Supplementary information:

Fig. 2. The frequencies should be given in THz, as in the rest of the manuscript.

Response:

We have changed Supplementary Figure 2 as requested.
Supplementary Figure 2 (Page 2).

Again, we thank Reviewer #1 for their constructive suggestions to improve our work.

Reviewer #2 (8):

The authors have substantially improved the presentation and essentially eliminated the concerns of my previous report.

They corrected the description of their magnetization and sublattice variables.

They clarified that they use an SU(3) theory (rather than SU(2)) in order to incorporate not only dipole but also quadrupole transitions. This seems to be needed here since the light field can induce $\Delta L = \pm 2$ (quadrupolar) transitions of the orbital part of the total angular momentum. This is an interesting aspect, now better explained in a new supplementary note.

They have extended the explanation of Fig. 1, although I am still somewhat doubtful about the readability of this figure.

Finally, and most importantly, the authors have performed an additional analysis of the output-to-input ratio, as suggested in my previous report. This analysis shows a substantially larger response of the CoO over other systems. I believe that this significantly enhances the importance of this paper and makes it overall suitable for Nature Communications.

I have been asked to comment also on the authors' response to the other reviewers' concerns. Some of these concerns coincide or have large overlap with my own previous concerns, as also recognized by the authors. I think that the authors thoroughly answered these concerns as well.

Therefore I can now recommend the paper for publication.

Response:

We thank Reviewer #2 for their positive evaluation of our work.

Reviewer #3 (12):

After reading the referee's reports and responses, the general remark from the previous round review is that the observation and experiments are potentially interesting but the original version of the manuscript has issues with presentation and insufficient discussion so that it was not acceptable for publication. In the revised manuscript, the authors have made important changes and improvement based on the suggestions from the reviewers. Most of the responses to my concerns are reasonable and satisfactory. However, there are still following questions that need to be addressed before I can recommend it for publication.

Response:

We thank Reviewer #3 for their positive evaluation of our work.

Reviewer #3 (13):

1. In the response to Reviewer #3 (11), I don't understand why the authors say "the magnon spectra at other pump polarization do not provide useful information". Wasn't the Fig.2c,d obtained by analyzing magnon spectra? Based on the time-domain traces shown in Fig.2a, their spectra should be quite different. For example, comparing the trace

at 94 degrees, the oscillation period measured at 4 degrees are much smaller. It means the high-frequency component should be dominant at 4 degrees. The authors should compare these spectra and discuss the difference.

Response:

We would like to explain that the results of the proposed analysis are already shown in Fig. 2c and d. In our analysis, from the time-domain traces (shown in Fig. 2a), we obtain magnon spectra through Fourier transformation of the traces. A particular example of the magnon spectrum at 94° is presented in Fig. 2b. For any (other) polarization, the magnon spectrum also exhibits three peaks, at around 4.4, 6.6 and 8.9 THz. We extracted the magnitudes of these three peaks as the most important information in these magnon spectra. The results of this peak analysis are shown in Fig. 2c and d. As the reviewer pointed out, the 4° time-domain trace is indeed dominated by the high-frequency component, which is observed as the almost zero amplitude of the 4.4-THz mode at $\theta = 4^\circ$ in Fig. 2c.

After receiving this comment, we realized that we should have stated “the magnon spectra at other pump polarization do not provide *additional* information,” because we meant that the essential information of the raw magnon spectra is already shown in Fig. 2c and d.

We believe that you should now find that Fig. 2c and d provide sufficient information for this discussion.

Reviewer #3 (14):

2. In supplementary figure 2a, authors assigned a peak at 200 cm⁻¹ to the 6.6 THz mode. However, there is another peak at 220 cm⁻¹, corresponding exactly to 6.6 THz mode. Why do the authors assign 220 cm⁻¹ peak to 6.6 THz mode? What is the origin of the 200 cm⁻¹ Raman peak?

Response:

We thank Reviewer #3 for pointing out this issue. According to the literature [15], the magnon peak should be at 220 cm⁻¹ (=6.6 THz) with a temperature-independent frequency and temperature-dependent intensity. Therefore, we have assigned the peak at 220 cm⁻¹ (=6.6 THz) to the 6.6-THz mode. This peak at 220 cm⁻¹ (=6.6 THz) was not visible in Supplementary Figure 2a, probably because the laser heating of up to 50 K diminished the intensity of this peak, as is seen in Supplementary Figure 1. The peak at 200 cm⁻¹ (=6.0 THz) was not found in any literature reports. Therefore, it may originate from impurities.

We added the above discussion to the Supplementary Figure as follows. Supplementary Figure 2 (Page 2).

Reviewer #3 (15):

3. In supplementary figure 2b, could author explain why the peak widths are much broader than the ones shown in Fig. 2b of the main text?

Response:

In Fig. 2b of the main text, the pump-probe measurement was performed at 5 K. In Supplementary Figure 2b, the spontaneous Raman scattering measurement was performed at 80 K, with estimated laser heating of 50 K. The higher temperature caused the difference in the peak widths.

We added the above discussion to the Supplementary Figure as follows.
Supplementary Figure 2 (Page 2).

Again, we thank Reviewer #3 for their constructive suggestions about our work.